psychology/behaviour

acoustics, voice gender, voice pitch, vocal tract resonances, gender development, masculinity

**Author for correspondence:**
Valentina Cartei
e-mail: v.cartei@sussex.ac.uk

# Children can control the expression of masculinity and femininity through the voice

Valentina Cartei, Alan Garnham, Jane Oakhill, Robin Banerjee, Lucy Roberts and David Reby

School of Psychology, University of Sussex, Brighton, UK

  VC, 0000-0001-8315-3595; AG, 0000-0002-0058-403X; DR, 0000-0001-9261-1711

Pre-pubertal boys and girls speak with acoustically different voices despite the absence of a clear anatomical dimorphism in the vocal apparatus, suggesting that a strong component of the expression of gender through the voice is behavioural. Initial evidence for this hypothesis was found in a previous study showing that children can alter their voice to sound like a boy or like a girl. However, whether they can spontaneously modulate these voice components within their own gender in order to vary the expression of their masculinity and femininity remained to be investigated. Here, seventy-two English-speaking children aged 6–10 were asked to give voice to child characters varying in masculine and feminine stereotypicality to investigate whether primary school children spontaneously adjust their sex-related cues in the voice—fundamental frequency ($F0$) and formant spacing ($\Delta F$)—along gender stereotypical lines. Boys and girls masculinized their voice, by lowering $F0$ and $\Delta F$, when impersonating stereotypically masculine child characters of the same sex. Girls and older boys also feminized their voice, by raising their $F0$ and $\Delta F$, when impersonating stereotypically feminine same-sex child characters. These findings reveal that children have some knowledge of the sexually dimorphic acoustic cues underlying the expression of gender, and are capable of controlling them to modulate gender-related attributes, paving the way for the use of the voice as an implicit, objective measure of the development of gender stereotypes and behaviour.

## 1. Introduction

Research on gender development consistently shows that children are aware of gender stereotypes, and this knowledge increases in range and complexity throughout childhood [1,2]. For example, as early as 2 years of age, boys and girls show awareness of concrete

items associated with their sex, such as which toys are for 'boys' or for 'girls' (e.g. dolls and trucks, [3] for a review). By fourth grade, children will spontaneously describe girls as nice, wearing dresses, and liking dolls, and boys as having short hair, playing active games and being rough with older children [4].

Although gender-typing is clearly multidimensional, past research has largely focused on the visual aspects of children's gender-typed knowledge and behaviour, such as physical appearance, choice of sex-typed activities or playmates, while little has been published on the ways in which attributes of the human voice are linked to gender development. This is surprising given that the human voice is highly sexually dimorphic from childhood: pre-pubertal boys speak with lower and more closely spaced vocal tract resonances (or formants) than females [5], while men also speak with lower fundamental frequency than women [6]. In adults, these acoustic differences are mainly determined by testosterone-driven changes in the male vocal apparatus during puberty [6,7]. Men develop longer vocal tracts than women, thus speaking with lower formants ($F_i$) and narrower formant spacing ($\Delta F$) compared to women, giving men's voices a more baritone quality. Men also grow longer and thicker vocal folds, therefore speaking on average with a lower mean fundamental frequency ($F0$), perceived as a lower pitch, than women. On the other hand, pre-pubertal sex differences have no clear anatomical origin: although $F0$ differences do not emerge until age 11 with the lack of pre-pubertal sex dimorphism in vocal fold length and mass [6], dimorphism in formant frequencies is apparent by about 4 years, despite no significant differences in vocal tract length between the two sexes before puberty [8–11].

More recently, Cartei and colleagues [12] have shown that children have knowledge of sexually dimorphic acoustic cues ($F0$ and $\Delta F$) and are capable of altering them to render vocal characteristics of their own or the opposite sex. They found that, when asked to imitate each of the two sexes, 6–9 year olds spontaneously lower their formants, thus narrowing their spacing, to sound more like a boy and raised them, thus widening their spacing, to sound more like a girl. Additionally, despite the confirmed absence of sex differences in the fundamental frequency of pre-pubertal children's natural voices, both boys and girls adjust this parameters when imitating the opposite sex, in line with the sex differences present in adults. These findings raise the interesting hypothesis that, as with other aspects of gendered behaviour, individuals acquire gender-typical ways of speaking from childhood. However, whether the control of sex-related acoustic parameters ($\Delta F$ and $F0$) extends to the expression of gendered characteristics (e.g. masculinity and femininity) remains to be investigated.

The present study seeks to address this gap in the literature by exploring the spontaneous ability of 6 to 10 year olds to modify the vocal expression of masculinity and femininity when giving voice to stereotypical and counter-stereotypical child characters of the same age and gender. We started at age 6 because the imitation task required children to read simple phrases. We stopped around age 10, because by 11–12 years of age the vocal tracts of children begin to show physical dimorphism [5]. We predicted that children would raise their $F0$ and widen $\Delta F$ when giving voice to stereotypically feminine characters, while they would lower their $F0$ and narrow $\Delta F$ when giving voice to stereotypically masculine characters.

# 2. Participants

A total of 72 children (36 girls), aged 6–10, took part in this study. Children were recruited from three primary schools. Twenty-five children were individually tested in a sound-attenuated room at the University campus, while the remainder were individually tested in a quiet room at their primary schools by two researchers. A three-stage consent procedure was employed. Head teachers first approved the study in each school and parental consent was then sought for invited participants. Participants provided their own verbal assent on the date of the study taking place. The procedure was granted ethics approval by the Sciences & Technology Cross-Schools Research Ethics Committee (C-REC) at the University of Sussex (Certificate: ER/VC44/16).

# 3. Methods

## 3.1. Audio recordings

Participants were seated in a comfortable chair and were audio recorded with a Zoom H1 handheld recorder, which was positioned at approximately 30 cm from the participant. A Marantz sound shield

surrounded the recorder to minimize environmental noise. Participants were instructed to keep as still as possible during the recordings, but could move freely otherwise.

Children were first audio recorded as they read out loud three sentences in their natural voice, 'Hello it is nice to meet you', 'Where were you yesterday?', 'No, I do not want to go', which were presented on paper. To overcome reading difficulties, children in Years 1 and 2 repeated the sentences after listening to an audio recording of an adult female voice saying the sentences. Next, all children performed the imitation task. In the task, they were presented with A4-sized descriptions of three fictional child characters of the same age and sex as the participant, but varying in masculine and feminine stereotypicality: the masculine character was described as having masculine interests and male friends, the feminine character as having feminine interests and female friends, and the gender-neutral character as having gender-neutral interests and male and female friends. This information was accompanied by a cartoon-style illustration of the target character's interests. The characters were presented one at a time and their order counter-balanced between participants.

After reading the information about a child character, participants were asked to repeat the same three sentences 'Hello it is nice to meet you', 'Where were you yesterday?', 'No, I do not want to go', as though they were that child (e.g. for the stereotypically masculine character the instructions read: 'Jacob is a boy. Jacob is your age. Jacob really likes playing with train sets and action toys. Jacob really likes playing with the boys in his class. Imagine you are Jacob. Jacob, can you repeat the following?'—see electronic supplementary material for the complete set of stimuli). Therefore, four recordings were obtained for each child, one recording encompassing the three sentences uttered in their natural voice, and three recordings (one per character type) encompassing the three sentences uttered as they gave voice to each of the three characters.

## 3.2. Acoustic analyses

Sound files were recorded at 44.1 kHz, 16 bits and saved in WAV format. Acoustic analyses were performed in Praat v.6.0.28 on a Mac [13]. The recordings were first edited manually to remove all silences, nonverbal vocalizations (e.g. laughter, loud breathing or nonverbal interjections) and the experimenter's questions, resulting in recordings that ranged in duration between 5.2 and 9.8 s (mean $8.0 \pm 1.1$ s). Children's mean $F0$ and first four formants $F_1 - F_4$ were then extracted from each recording using a customized script in PRAAT for batch processing (available on request).

The script calculated fundamental frequency ($F0$) using the PRAAT autocorrelation algorithm 'to Pitch' with a pitch floor 100 Hz and ceiling 600 Hz, time step 0.01 s, and the formants $F_1 - F_4$ using Praat's Burg linear predictive coding algorithm with the initial settings of maximum formant 8000 Hz and formant number 5, dynamic range 30 dB, length of the analysis window 0.3 s. Formant values were also overlaid on a spectrogram and maximum formant and formant number parameters were manually adjusted until the best visual fit of predicted onto observed formants was obtained. Formant $F_1 - F_4$ values of each recording were then used to derive average formant spacing ($\Delta F$), which is average distance between any two adjacent formants ($\Delta F = F_{i+1} - F_i$) (full details of this method are given in electronic supplementary material, appendix S1). Hence, the longer the vocal tract, the lower the formant frequencies, and the narrower their overall frequency spacing. Because $\Delta F$ is expressed in Hz, we also report the apparent Vocal Tract Length (aVTL), the inverse acoustic correlate of $\Delta F$, which is expressed in cm. Apparent Vocal Tract Length provides an estimate of 'speaking' VTL, the anatomical vocal tract length achieved during phonation (as opposed to 'resting' VTL, which is the anatomical VTL achieved during quiet breathing [14]).

# 4. Results

## 4.1. Age and sex differences in the natural voice

We performed a series of ANCOVAs to test the effects of sex (independent variable) and age (continuous covariate) on the acoustic parameters $F0$ and $\Delta F$ of children's natural voices. Simple linear regressions were performed on the boys' and girls' data separately to further examine the relationship between the acoustic parameters and age within each sex. There was a significant effect of the age covariate on mean $F0$, $F_{1,72} = 13.06$, $p = 0.001$, and $\Delta F$, $F_{1,72} = 19.46$, $p < 0.001$. Simple regressions (with 1000 bootstrap samples) showed that, as children get older, both sexes speak with a lower $F0$ (girls: $R^2 = 0.13$, $F_{1,34} = 5.08$, $\beta = -0.36$, $p = 0.031$, bootstrap 95% CI: $-10.19$, $-0.68$, figure 1a; boys: $R^2 = 0.21$,

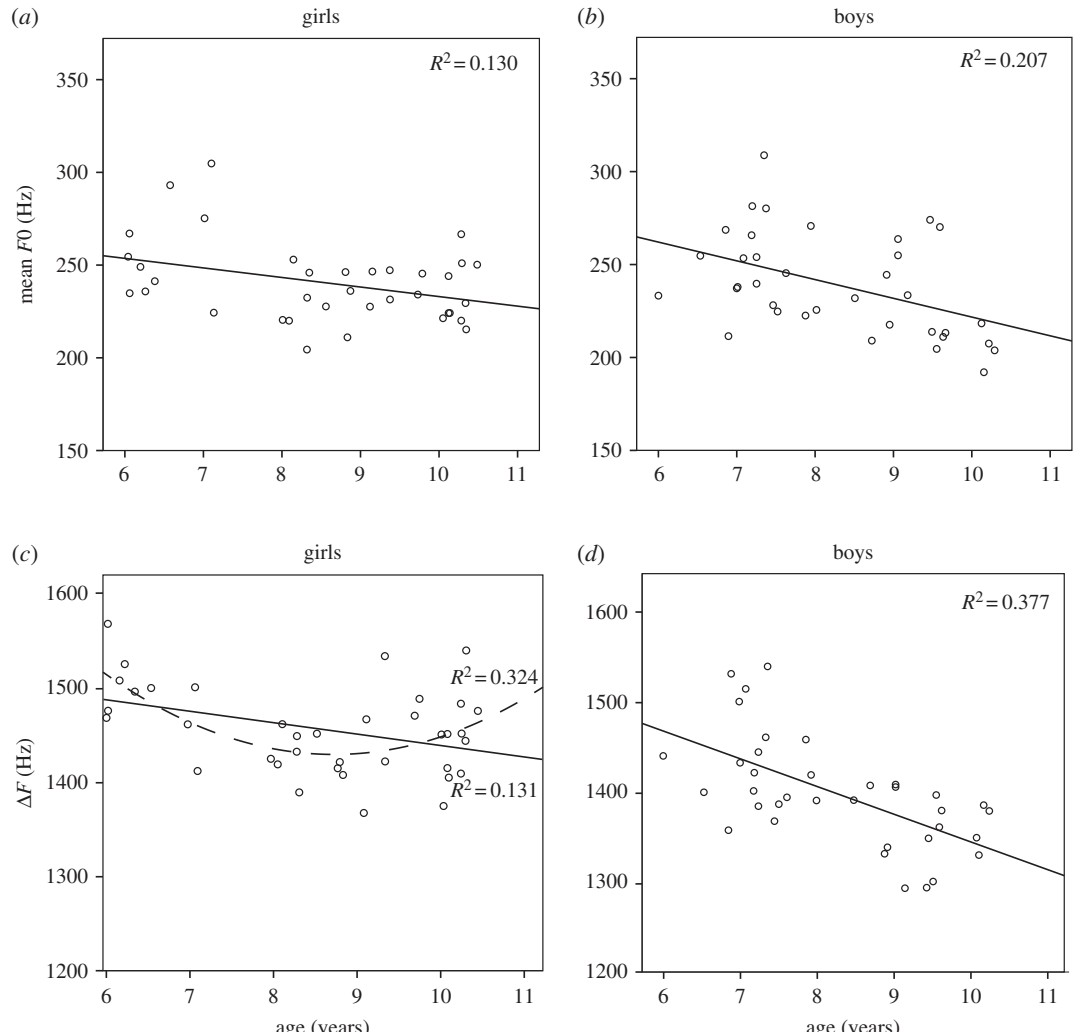

**Figure 1.** Scatterplots of mean fundamental frequency (mean *F*0 in Hz) and formant spacing (Δ*F* in Hz) in girls' and boys' natural speaking voices.

$F_{1,34} = 8.85$, $\beta = -0.46$, $p = 0.005$, bootstrap 95% CI: $-16.42$, $-3.19$, figure 1*b*) and a narrower Δ*F* (girls: $R^2 = 0.11$, $F_{1,34} = 4.22$, $\beta = -0.33$, $p = 0.048$, bootstrap 95% CI: $-19.96$, $-0.11$, figure 1*c*; boys: $R^2 = 0.38$, $F_{1,34} = 20.58$, $\beta = -0.61$, $p < 0.001$, bootstrap 95% CI: $-43.8$, $-0.17.8$, figure 1*d*). However, in girls the relationship between Δ*F* and age was better described by a quadratic model ($R^2 = 0.32$, $F_{1,34} = 8.145$, $\beta = -6.45$, $p = 0.001$) rather than the linear model ($R^2 = 0.32$, $F_{1,34} = 5.27$, $\beta = -0.362$, $p = 0.028$), with a downward peak in girls' Δ*F* between the ages of 8 and 9.

There were no significant sex differences in children's mean *F*0, $p\text{s} > 0.05$ (girls: $M$(s.e.) $= 241.44(3.79)$, boys: $M$(s.e.) $= 238.103(3.79)$). However, the effect of sex on children's natural Δ*F* was significant: boys spoke with a 63.85 Hz (0.54 cm) lower Δ*F* ($M$(s.e.) $= 1396.7(8.0)$ Hz or aVTL $= 12.53$ cm) than girls ($M$(s.e.) $= 1460.5(8.0)$Hz or aVTL $= 11.97$ cm), $F_{1,72} = 31.43$, $p < 0.001$. The age by sex interaction was significant, with the magnitude of the between-sex difference in Δ*F* increasing with age, $F_{1,68} = 6.23$, $p = 0.015$.

## 4.2. Ability to control voice gender

We assessed the ability of boys and girls to shift *F*0 and Δ*F* by testing the main effects of character type (as a three-level within-subject factor: feminine, gender-neutral, masculine) on these acoustic parameters with repeated measures ANCOVAs for boys and girls separately (scatterplots of residuals are reported as electronic supplementary material, figures S1 and S2). Age was mean centred before being entered in the

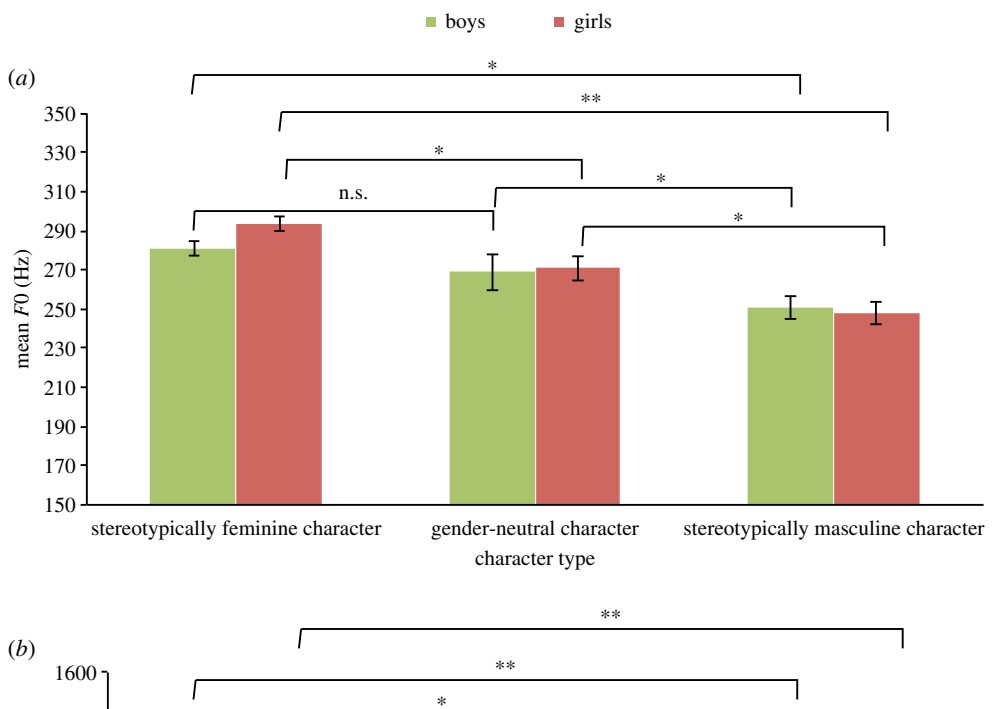

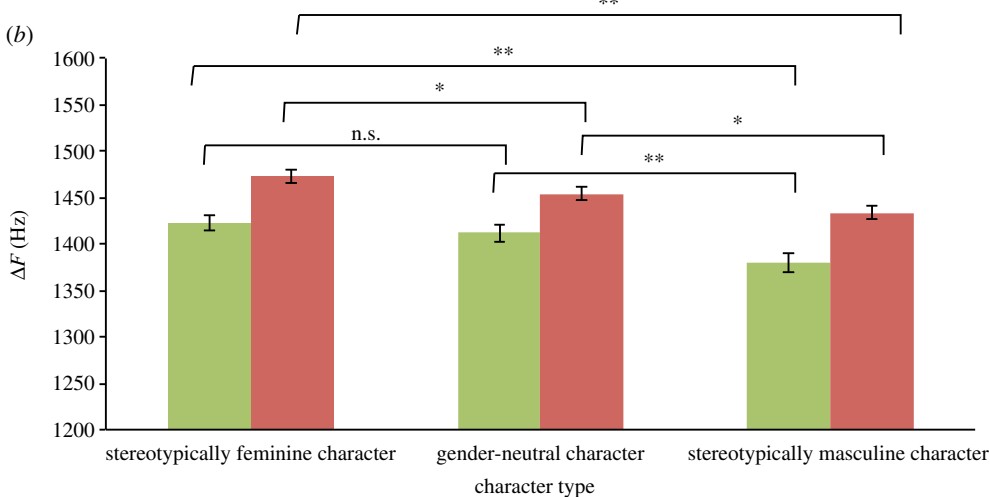

**Figure 2.** Bar graphs representing $F0$ and $\Delta F$ of boys' and girls' voices for each character type (feminine, gender-neutral, masculine).

model as a covariate, to prevent the covariate from altering the main effect of the repeated measure [15]. All pairwise comparisons were Bonferroni corrected.

Means and s.e. (as error bars) of $F0$ and $\Delta F$ for each character type are shown in figure 2 (also in electronic supplementary material, figure S3: violin plots), along with significance from pairwise comparisons between the three character types (see electronic supplementary material, table S1 for $F0$ and $\Delta F$ descriptives). Significant interactions of age by character type were further investigated by calculating the difference in $F0$ and $\Delta F$ between the gender-neutral and masculinized or feminized imitations and regressing these difference variables on age.

### 4.2.1. Fundamental frequency ($F0$)

The main effect of character type on mean $F0$ was significant in both girls, $F_{2,68} = 24.61, p < 0.001$, and boys, $F_{2,68} = 8.68, p < 0.001$. Pairwise comparisons (figure 2a) revealed that girls and boys significantly lowered their $F0$ by 22.7 Hz (8.4%) and 17.9 Hz (7.1%) respectively, when giving voice to masculine characters compared to the gender-neutral characters, and by 45.61 Hz (15.5%) and by 30.2 Hz (12%) compared to the feminine characters, all $p$s < 0.05. Girls also significantly raised their $F0$ by 22.9 (8.5%) Hz when giving voice to the feminine character, compared to gender-neutral character, $p = 0.015$. Boys' upward shift in $F0$ of 12.4 Hz (4.6%) for feminine characterizations was not significant, $p > 0.05$.

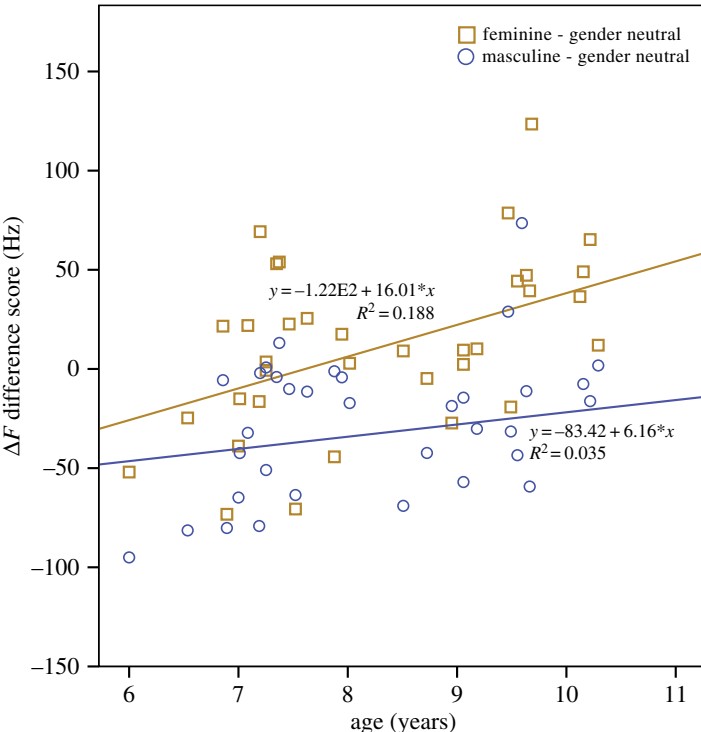

**Figure 3.** Linear regressions of boys' difference $\Delta F$ values (feminine minus gender-neutral $\Delta F$ values, gender-neutral minus masculine $\Delta F$ values) over age.

There was a significant effect of age as the covariate on boys' mean $F0$, $F_{1,34} = 5.81$, $p = 0.022$ with mean $F0$ across character types linearly decreasing with boys' age. No significant effect of age on girls' mean $F0$ was found, $p > 0.05$. No significant interactions between character type and age on children's $F0$ were found.

### 4.2.2. Formant spacing ($\Delta F$)

The main effect of character type on $\Delta F$ was significant in both girls, $F_{2,68} = 27.62$, $p < 0.001$, and boys, $F_{2,68} = 24.81$, $p < 0.001$. Girls and boys significantly narrowed their $\Delta F$ by 20.30 Hz (0.17 cm) and 32.4 Hz (0.29 cm) respectively when giving voice to the masculine characters compared to the gender-neutral characters, and by 39.48 Hz (0.32 cm) and 43.13 Hz (0.38 cm) compared to the feminine characters, all $p$s < 0.05. Girls also significantly widened their $\Delta F$ by 20 Hz (0.16 cm) for the feminine character, compared to the gender-neutral character (figure 2b), $p = 0.009$. Boys' upward shift in $\Delta F$ of 10.7 Hz (0.09 cm) for the feminine character compared to the gender-neutral character was not significant, $p > 0.05$.

There was a significant effect of age as the covariate on $\Delta F$ in both sexes, with a narrowing of $\Delta F$ across all character types as children got older, (girls: $F_{1,34} = 13.60$, $p < 0.001$; boys: $F_{1,34} = 10.34$, $p = 0.003$). There was a significant interaction effect of age with character type in boys only, $F_{2,68} = 4.57$, $B = 0.43$, $p = 0.014$. Simple regressions (with 1000 bootstrap samples) revealed that the magnitude of the difference in $\Delta F$ between boys' imitations of the feminine and gender-neutral character types significantly increased with participant age, $R^2 = 0.18$, $F_{1,34} = 7.90$, $\beta = 0.43$, $p = 0.008$, bootstrap 95% CI: 5.70, 27.13 (figure 3). Age was not significantly related to the magnitude of the difference in $\Delta F$s between boys' imitations of the gender-neutral and masculine characters, $\beta = 0.04$, $p > 0.05$, bootstrap 95% CI: −5.75, 17.39.

## 5. Discussion

This study shows that from at least 6 years of age, children spontaneously vary the masculinity and femininity of their voice for gender stereotypical and counter-stereotypical characterizations, suggesting that children have an awareness and ability to control the vocal as well as visual aspects of

gendered expression (e.g. physical appearance, play styles, toy preferences: see [16,17]). Below, we discuss the potential role of vocal behaviour in signalling gender and associated attributes in relation to the between-sex differences observed in children's natural voices and the within-sex differences observed in the imitation task.

When analysing children's natural speaking voices, we found age and sex differences that are consistent with published data on $F0$ and $\Delta F$ of English-speaking children ([5] for a review). More specifically, we found that, while there were no sex differences in $F0$, boys spoke with an overall 4.5% lower $\Delta F$ than girls, and the magnitude of this difference increased with age (up to 10% by age 10). As vocal tract length scales with overall body growth, we also found a steady and linear decrease of $\Delta F$ in boys from age 6 to 10 by approximately 22%, in line with published increases in body height for this age range (approx. 20%—[7]). Contrary to anatomical vocal tract growth, however, girls' $\Delta F$ followed a quadratic relationship with $\Delta F$ increasing again after age 9. While a larger dataset is needed to confirm these findings, our observations are consistent with the assumption that the overall pre-pubertal dimorphism in formant frequency values is a consequence of sex-specific behaviours [18], such as girls speaking with spread lips ('with a smile'), which would in turn shorten their tracts [6,19]. Interestingly, a large facial study examining gender differences in smiling behaviour by using yearbook photos, has shown that between the ages of 9 and 12, girls begin to smile more than boys in photographs and this difference persists well into adulthood [20]. While several authors have suggested that sex differences in smiling are based on gender role expectations e.g. women being more 'gentle', 'unthreatening' and 'empathic' than men [21,22], the interactions between facial and vocal behaviours in the gender expression of adults and children remain to be systematically investigated.

Turning to the imitation task, overall our results support the hypothesis that children are capable of varying the expression of masculinity and femininity through their voice from at least 6 years of age (the youngest children in our sample). Both sexes lowered the $F0$ and $\Delta F$ of their voices when imitating a stereotypically masculine child character and no appreciable differences were found in children's masculine impersonations with age. Both sexes were also capable of raising $F0$ and $\Delta F$ to feminize their voices for the stereotypically feminine child characters, although boys' upward shifts in these acoustic parameters did not reach overall significance (a result that could be further investigated by replicating the study on a larger sample). Interestingly, while girls' performance for the feminine characterizations did not change with age, a significant widening of $\Delta F$ with age was observed for boys' imitations of the feminine character type.

The observed sex differences in children's imitations of feminized characters are very unlikely to originate from anatomical or mechanistic constraints in boys' vocal production, given that, when asked to imitate the other sex, $F0$ and $\Delta F$ values of boys aged 6–9 were similar to those of girls in the same age range [12]. Moreover, both sexes are capable of masculinizing and feminizing their voice in adulthood, once sex differences in the anatomy of the vocal apparatus are actually present [14].

It is also unlikely that these differences in vocal feminization reflect gender differences in accessibility of stereotype domains, as the feminine characterizations used in this study are the same as the ones used in a previous psychoacoustic study [23], which showed that 7- and 8-year-old boys and girls similarly preferentially attributed feminized boy voices over masculinized voices to feminized boy characters. Moreover, research on toy/activity preferences reports that from kindergarten boys and girls show a similar age-related increase in knowledge and use of appearance, activity and trait stereotypes for boy targets [4].

One more likely explanation relates to the fact that gender expression is understood differently by the two sexes. Miller and colleagues [4] have recently shown that, when asked to describe boys and girls, children, and particularly boys, tend to define girls by appearance, while boys are more likely to be described in terms of their activities and interests. Consequently, for the younger boys in our sample a boy character who engages in female stereotyped activities would still possess masculine appearance attributes, including their voice, in order to be a 'boy'.

It is possible that the older boys in our study may have felt more comfortable in displaying feminine behaviour compared to the younger ones. Indeed, children, and particularly boys, become more flexible with age in terms of making counter-stereotypical associations with their gender [16]. Moreover, most research agrees that younger (4–7 year olds) children, and particularly younger boys, are more inflexible than girls in their judgements of gender norm violations [24], and less likely to act in counter-stereotypical ways (e.g. girls are more likely to play with boys' toys and wear boys' clothes than boys are likely to play with girls' toys and wear girls' clothes [17,25,26]). To investigate these hypotheses, replications of the present study could include measures of children's sex typing of themselves and others (e.g. see [27] for a review of these measures).

While this study shows that children control their voices in a way that accentuates or downplays gender attributes, at a perceptual level listeners are likely to be affected by these behavioural adjustments when characterizing speakers' masculinity and femininity. Indeed, psychoacoustic studies have shown that adult listeners are sensitive to artificial manipulations of $F0$ and $\Delta F$, with lower-pitched (in adults) and more resonant voices (in both children and adults) being consistently rated as more masculine than their higher-pitched, less resonant counterparts [28–30]. However, no study has so far investigated whether adjustments of $F0$ and $\Delta F$ in child speakers' natural voices also have a perceptual relevance in gendered attributions made by child and adult listeners. Future studies could also explore whether such judgements relate to child speakers' gendered characteristics in other domains (e.g. appearance, gait), as well as to listeners' attributions of likability, as children who deviate from gender voice stereotypes may incur peer rejection, especially in contexts and cultures where gender norms are particularly salient.

Our results also raise questions about the potential impact of voice gender stereotyping in the context of peer relationships. Peer group interactions have long been recognized as important for the development of gendered behaviours [31], and interest in conforming to gender stereotypes in the presence of same-sex peer groups has been shown to be one important example of early self-presentational behaviour [16]. Coupling this evidence with our present findings that children are able to manipulate their voices systematically to sound more masculine or feminine, provides a valuable opportunity to test hypotheses about how children deploy vocal strategies in order to convey information about sex-typed attributes to peers.

Finally, we propose that the voice imitation paradigm used in this study could provide an implicit measure of children's stereotyping at a younger age than what is achievable by current methods based on response latency techniques. These methods are routinely used in adults and children of school age, yet the cognitive demands of these procedures make their use with pre-schoolers more difficult (e.g. IAT— [32,33]). Given that we can quantify changes in sex-related acoustic parameters, and that even very young children engage in role play [34,35], the present imitation task could be easily adapted to be independent of reading ability and thus used with younger children, for example by replacing the reading of sentences with a recording of simple sentences or words, that children would have to repeat. Thus, voice production tasks such as the one in the present study are a highly adaptable and viable method of accessing the development of gender stereotypes, and associated unconscious biases in children and adults.

Ethics. The study was reviewed and approved by the Sciences and Technology Cross-Schools Research Ethics Committee (C-REC) of the University of Sussex (code: ER/VC44/14).

Permission to carry out fieldwork. Permission to conduct the study in the schools was obtained via C-REC (code: ER/VC44/14), with the head teachers' and parents' consent.

Data accessibility. The dataset supporting this article has been uploaded to the Sussex Research Online (SRO) repository: https://sussex.figshare.com/s/285aef9e3bd234a5541c

Authors' contributions. V.C. participated in the data collection, contributed to the design, performed acoustic and statistical analyses, wrote the manuscript and created the figures; R.B., A.G., J.O. contributed to the design of the investigation; L.R. collected field data and participated in the acoustic analyses; D.R. conceived of the study, participated in the design and analyses, and helped draft the manuscript. The manuscript was reviewed, edited and approved by all authors, who agree to be accountable for the work.

Competing interests. The authors declare no competing interests.

Funding. The Leverhulme Trust funded this research (grant no. RPG-2016-396).

Acknowledgements. We are grateful to the children and their parents who agreed to take part in the study, and to the head teachers and staff of the schools who participated. We are grateful to Dr Sophie Anns for her assistance with the data collection. The Leverhulme Trust funded this research (grant no. RPG-2016-396).

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
