## [Reviewer comments · Royal Society Open Science]

Review History

RSOS-190656.R0 (Original submission)

Review form: Reviewer 1

Is the manuscript scientifically sound in its present form?

Yes

Are the interpretations and conclusions justified by the results?

Yes

Is the language acceptable?

Yes

Is it clear how to access all supporting data?

Yes

Do you have any ethical concerns with this paper?

No

Have you any concerns about statistical analyses in this paper?

Yes

Recommendation?

Major revision is needed (please make suggestions in comments)

Comments to the Author(s)

This is a very interesting, well written and coherent article, that directly tackles an important gap in the literature. It reports the results of a study investigating whether children can modulate their voice parameters (namely, F0 and ΔF) to modulate gender-related attributes when impersonating same-sex child characters. The results suggest that, in fact, both girls and boys lower F0 and ΔF to masculinise their voices when impersonating stereotypically masculine child characters, and (girls and older boys) raise them for stereotypically feminine same-sex child characters. This is particularly interesting, as the authors mentioned, given the strong sexual dimorphism of the human voice, and the little attention it has received as a marker of the development of gender stereotypes.

The results are sensible, but the manuscript has a few problems (detailed below) that should be addressed before publication. With some modification the manuscript would be appropriate for publication.

General comments:

The main (but solvable) problem I find is related to the statistics. Given that this is a novel (and somewhat exploratory) study, it is understandable that there is not a priori power (and sample size) analysis. However, this creates a problem for the interpretation of statistical significance, as the existence of an association cannot be inferred simply from p values. In other words, the study may be, for example, underpowered to detect some moderate or weak associations between predictors and outcome variables. In addition, there is no information of influential data points, nor diagnostics of the adequacy model fit (i.e. residual distribution for ANCOVAs and regressions, which I suggest could be included as supplemental material). I would also recommend providing additional information to infer potential associations, like using bootstrap techniques to identify confidence intervals (having CIs would help assessing associations even in the absence of p values). In particular, bootstrapping regressions could really help in dealing with all these issues (see e.g. Fox, 2016, Chapter 21).

In addition, there are few descriptive statistics reported, and I think these are important for interpretation and to inform power calculations in future studies and potential replication attempts. I recommend the authors to include a table of descriptives (e.g. n, mean, median, SD, min, max) of F0 and ΔF for the impersonations of the three characters, by participant sex and age.

Also, according to the review system "It is a condition of publication that authors make their supporting data, code and materials available", but the link the authors provided to access the data (<http://sro.sussex.ac.uk/id/eprint/83066/>) gives a "404 File not Found" error. Also, I did not see any mention of code (or SPSS syntax, which from the figures I think the authors used) to reproduce the analyses, but because these are clearly described in the manuscript, it is perhaps desirable, but not too necessary in this particular case to include it (other supplementary files are available with the submitted files).

Finally, perhaps due to some sort of problem or confusion with the submission system, or an author oversight, there are no figure captions.

Specific comments:

Page 2, line 52. The citation should be “Perry, Ohde, & Ashmead, 2001”, instead of “Perry & Ashmead, 2001”.

Page 3, lines 73-76. In the Participants section, the authors categorise the ages of the participant children in years 1 to 6. I wonder why use Years 1-6, instead of age 6-10? I'd rather use the actual age for the sake of simplicity. It is not clear (at least for me) why these categories are used and, more importantly, why use 6 categories when there are only 5 years of age under study. However, if there is a justification behind this decision, I would like to see it in the manuscript, because it is not obvious to me (and probably some readers of the published paper).

Page 3, line 80. “Participants provided their own verbal consent”. Given that they were children, very young in some cases, shouldn't it be “assent” instead of “consent”? (see De Lourdes Levy, Larcher, & Kurz, 2003).

Page 4, line 143. For consistency, I think the first results reported in “F0 ($R^2 = .13$, $F(1, 34) = 5.08$, $\beta = -0.36$, $p = .031$, Figure 1a; boys: $R^2 = .21$, $F(1, 34) = 8.89$, $\beta = -.46$, $p = .005$, Figure 1b)” should start with “girls:”.

Page 5, line 145. Figure 2a cited here, only illustrates the quadratic model, described below without reference to the figure, but not the linear one reported here. Perhaps both the linear and quadratic could be shown in this same figure panel?

Page 5, lines 153-154. This is not necessary, but I would personally like to see the significant age and sex interaction here reported illustrated with a figure (or figure panel). Also, Figs 1 and 2 could perhaps be just one fig with 4 (or 5) panels, illustrating all results related to Age and sex differences in the natural voice.

Page 5, lines 159-160. The authors report that the covariate (age) was men centred. This is the correct decision, but when the interaction is significant, I think the authors should centre to at least low, high and central (age) values to explore the interaction between character type and age (see Delaney & Maxwell, 1981; Schneider, Avivi-Reich, & Mozuraitis, 2015; Taylor, 2011; for an example of this in an analogous design on voice research, see Leongómez, Mileva, Little, & Roberts, 2017). I really do think this will help interpret the interaction, and how the model slopes vary for kids of different ages. In addition, this technique will keep the n as the values are estimated, and you would not lose statistical power.

Page 5, line 163. Did you adjust the alpha for multiple comparisons? An alternative, to limit Type I error inflation, would be to instead use (simple) planned contrasts on the estimated values from the covariate (centred to different values), using the neutral character responses as reference (i.e. comparing responses to the feminine and masculine character vs the neutral). In SPSS, see https://www.ibm.com/support/knowledgecenter/no/SSLVMB_24.0.0/spss/advanced/syn_glm_repeated_measures_contrasts_wsfactor.html

Page 6, line 192. Here I would suggest centring to low, mid, and high age.

Page 6, lines 210-211. “(...) boys spoke with an overall 4.5% lower ΔF in girls” should be “4.5% lower ΔF than girls”.

References

- De Lourdes Levy, M., Larcher, V., & Kurz, R. (2003). Informed consent/assent in children. Statement of the Ethics Working Group of the Confederation of European Specialists in Paediatrics (CESP). *European Journal of Pediatrics*, 162(9), 629–633.
<https://doi.org/10.1007/s00431-003-1193-z>
- Delaney, H. D., & Maxwell, S. E. (1981). On Using Analysis Of Covariance In Repeated Measures Designs. *Multivariate Behavioral Research*, 16(1), 105–123.
https://doi.org/10.1207/s15327906mbr1601_6
- Fox, J. (2016). *Applied Regression Analysis and Generalized Linear Models* (3rd ed.). Thousand Oaks, CA: Sage. Retrieved from https://us.sagepub.com/sites/default/files/upm-binaries/68018_Fox_Chapter_21.pdf
- Leongómez, J. D., Mileva, V. R., Little, A. C., & Roberts, S. C. (2017). Perceived differences in social status between speaker and listener affect the speaker's vocal characteristics. *PLoS One*, 12(6), e0179407. <https://doi.org/10.1371/journal.pone.0179407>
- Schneider, B. A., Avivi-Reich, M., & Mozuraitis, M. (2015). A cautionary note on the use of the Analysis of Covariance (ANCOVA) in classification designs with and without within-subject factors. *Frontiers in Psychology*, 6. <https://doi.org/10.3389/fpsyg.2015.00474>
- Taylor, A. (2011). *Using the GLM Procedure in SPSS*. Sydney, Australia: Macquarie University. Retrieved from <http://www.psy.mq.edu.au/psystat/documents/GLMSPSS.pdf>

Review form: Reviewer 2

Is the manuscript scientifically sound in its present form?

Yes

Are the interpretations and conclusions justified by the results?

Yes

Is the language acceptable?

Yes

Is it clear how to access all supporting data?

Yes

Do you have any ethical concerns with this paper?

No

Have you any concerns about statistical analyses in this paper?

No

Recommendation?

Accept with minor revision (please list in comments)

Comments to the Author(s)

This is an interesting paper reporting a straightforward study that shows further evidence that pre-pubertal children are aware of the stereotypes associated with voice pitch. My only suggestion is to improve the figures. Violin plots would be a much better way to present the data in the bar charts, for example.

Decision letter (RSOS-190656.R0)

03-Jun-2019

Dear Dr Cartei

On behalf of the Editors, I am pleased to inform you that your Manuscript RSOS-190656 entitled "Children can control the expression of masculinity and femininity through the voice" has been accepted for publication in Royal Society Open Science subject to minor revision in accordance with the referee suggestions. Please find the referees' comments at the end of this email.

The reviewers and handling editors have recommended publication, but also suggest some minor revisions to your manuscript. Therefore, I invite you to respond to the comments and revise your manuscript.

- Ethics statement

- Data accessibility

If you wish to submit your supporting data or code to Dryad (<http://datadryad.org/>), or modify your current submission to dryad, please use the following link:
<http://datadryad.org/submit?journalID=RSOS&manu=RSOS-190656>

- Competing interests

- Authors' contributions

All submissions, other than those with a single author, must include an Authors' Contributions section which individually lists the specific contribution of each author. The list of Authors should meet all of the following criteria; 1) substantial contributions to conception and design, or

acquisition of data, or analysis and interpretation of data; 2) drafting the article or revising it critically for important intellectual content; and 3) final approval of the version to be published.

- Acknowledgements

- Funding statement

Because the schedule for publication is very tight, it is a condition of publication that you submit the revised version of your manuscript before 12-Jun-2019. Please note that the revision deadline will expire at 00.00am on this date. If you do not think you will be able to meet this date please let me know immediately.

- 1) A text file of the manuscript (tex, txt, rtf, docx or doc), references, tables (including captions) and figure captions. Do not upload a PDF as your "Main Document";

- 2) A separate electronic file of each figure (EPS or print-quality PDF preferred (either format should be produced directly from original creation package), or original software format);
- 3) Included a 100 word media summary of your paper when requested at submission. Please ensure you have entered correct contact details (email, institution and telephone) in your user account;
- 4) Included the raw data to support the claims made in your paper. You can either include your data as electronic supplementary material or upload to a repository and include the relevant doi within your manuscript. Make sure it is clear in your data accessibility statement how the data can be accessed;
- 5) All supplementary materials accompanying an accepted article will be treated as in their final form. Note that the Royal Society will neither edit nor typeset supplementary material and it will be hosted as provided. Please ensure that the supplementary material includes the paper details where possible (authors, article title, journal name).

on behalf of Professor Carolyn McGettigan (Associate Editor) and Essi Viding (Subject Editor)
openscience@royalsociety.org

Associate Editor Comments to Author (Professor Carolyn McGettigan):

I have now received 2 reviews of your paper from relevant experts in the field. I'm happy to report that both reviewers find the study to be of great interest. As you will see, their queries mainly concern some aspects of the reporting of the statistical analyses, presentation of the results, and the availability of the supporting data and code. Please take a look at their requests

and address each of them with a revised manuscript - my view is that they constitute a somewhat minor revision to the paper. If any of the requested adjustments to the statistics (e.g. corrections for multiple comparisons) should alter the findings substantially, then please do highlight this in your response letter and discuss accordingly in the revised manuscript.

Reviewer comments to Author:

Reviewer: 1

Comments to the Author(s)

This is a very interesting, well written and coherent article, that directly tackles an important gap in the literature. It reports the results of a study investigating whether children can modulate their voice parameters (namely, F0 and ΔF) to modulate gender-related attributes when impersonating same-sex child characters. The results suggest that, in fact, both girls and boys lower F0 and ΔF to masculinise their voices when impersonating stereotypically masculine child characters, and (girls and older boys) raise them for stereotypically feminine same-sex child characters. This is particularly interesting, as the authors mentioned, given the strong sexual dimorphism of the human voice, and the little attention it has received as a marker of the development of gender stereotypes.

The results are sensible, but the manuscript has a few problems (detailed below) that should be addressed before publication. With some modification the manuscript would be appropriate for publication.

General comments:

The main (but solvable) problem I find is related to the statistics. Given that this is a novel (and somewhat exploratory) study, it is understandable that there is not a priori power (and sample size) analysis. However, this creates a problem for the interpretation of statistical significance, as the existence of an association cannot be inferred simply from p values. In other words, the study may be, for example, underpowered to detect some moderate or weak associations between predictors and outcome variables. In addition, there is no information of influential data points, nor diagnostics of the adequacy model fit (i.e. residual distribution for ANCOVAs and regressions, which I suggest could be included as supplemental material). I would also recommend providing additional information to infer potential associations, like using bootstrap techniques to identify confidence intervals (having CIs would help assessing associations even in the absence of p values). In particular, bootstrapping regressions could really help in dealing with all these issues (see e.g. Fox, 2016, Chapter 21).

In addition, there are few descriptive statistics reported, and I think these are important for interpretation and to inform power calculations in future studies and potential replication attempts. I recommend the authors to include a table of descriptives (e.g. n, mean, median, SD, min, max) of F0 and ΔF for the impersonations of the three characters, by participant sex and age.

Also, according to the review system "It is a condition of publication that authors make their supporting data, code and materials available", but the link the authors provided to access the data (<http://sro.sussex.ac.uk/id/eprint/83066/>) gives a "404 File not Found" error. Also, I did not see any mention of code (or SPSS syntax, which from the figures I think the authors used) to reproduce the analyses, but because these are clearly described in the manuscript, it is perhaps desirable, but not too necessary in this particular case to include it (other supplementary files are available with the submitted files).

Finally, perhaps due to some sort of problem or confusion with the submission system, or an author oversight, there are no figure captions.

Specific comments:

Page 2, line 52. The citation should be “Perry, Ohde, & Ashmead, 2001”, instead of “Perry & Ashmead, 2001”.

Page 3, lines 73-76. In the Participants section, the authors categorise the ages of the participant children in years 1 to 6. I wonder why use Years 1-6, instead of age 6-10? I'd rather use the actual age for the sake of simplicity. It is not clear (at least for me) why these categories are used and, more importantly, why use 6 categories when there are only 5 years of age under study. However, if there is a justification behind this decision, I would like to see it in the manuscript, because it is not obvious to me (and probably some readers of the published paper).

Page 3, line 80. “Participants provided their own verbal consent”. Given that they were children, very young in some cases, shouldn't it be “assent” instead of “consent”? (see De Lourdes Levy, Larcher, & Kurz, 2003).

Page 4, line 143. For consistency, I think the first results reported in “F0 ($R^2 = .13$, $F(1, 34) = 5.08$, $\beta = -0.36$, $p = .031$, Figure 1a; boys: $R^2 = .21$, $F(1, 34) = 8.89$, $\beta = -.46$, $p = .005$, Figure 1b)” should start with “girls: ”.

Page 5, line 145. Figure 2a cited here, only illustrates the quadratic model, described below without reference to the figure, but not the linear one reported here. Perhaps both the linear and quadratic could be shown in this same figure panel?

Page 5, lines 153-154. This is not necessary, but I would personally like to see the significant age and sex interaction here reported illustrated with a figure (or figure panel). Also, Figs 1 and 2 could perhaps be just one fig with 4 (or 5) panels, illustrating all results related to Age and sex differences in the natural voice.

Page 5, lines 159-160. The authors report that the covariate (age) was men centred. This is the correct decision, but when the interaction is significant, I think the authors should centre to at least low, high and central (age) values to explore the interaction between character type and age (see Delaney & Maxwell, 1981; Schneider, Avivi-Reich, & Mozuraitis, 2015; Taylor, 2011; for an example of this in an analogous design on voice research, see Leongómez, Mileva, Little, & Roberts, 2017). I really do think this will help interpret the interaction, and how the model slopes vary for kids of different ages. In addition, this technique will keep the n as the values are estimated, and you would not lose statistical power.

Page 5, line 163. Did you adjust the alpha for multiple comparisons? An alternative, to limit Type I error inflation, would be to instead use (simple) planned contrasts on the estimated values from the covariate (centred to different values), using the neutral character responses as reference (i.e. comparing responses to the feminine and masculine character vs the neutral). In SPSS, see https://www.ibm.com/support/knowledgecenter/no/SSLVMB_24.0.0/spss/advanced/syn_glm_repeated_measures_contrasts_wsfactor.html

Page 6, line 192. Here I would suggest centring to low, mid, and high age.

Page 6, lines 210-211. “(...) boys spoke with an overall 4.5% lower ΔF in girls” should be “4.5% lower ΔF than girls”.

References

De Lourdes Levy, M., Larcher, V., & Kurz, R. (2003). Informed consent/assent in children. Statement of the Ethics Working Group of the Confederation of European Specialists in Paediatrics (CESP). *European Journal of Pediatrics*, 162(9), 629–633.
<https://doi.org/10.1007/s00431-003-1193-z>

Delaney, H. D., & Maxwell, S. E. (1981). On Using Analysis Of Covariance In Repeated Measures Designs. *Multivariate Behavioral Research*, 16(1), 105–123.
https://doi.org/10.1207/s15327906mbr1601_6

Fox, J. (2016). *Applied Regression Analysis and Generalized Linear Models* (3rd ed.). Thousand Oaks, CA: Sage. Retrieved from https://us.sagepub.com/sites/default/files/upm-binaries/68018_Fox_Chapter_21.pdf

Leongómez, J. D., Mileva, V. R., Little, A. C., & Roberts, S. C. (2017). Perceived differences in social status between speaker and listener affect the speaker’s vocal characteristics. *PLoS One*, 12(6), e0179407. <https://doi.org/10.1371/journal.pone.0179407>

Schneider, B. A., Avivi-Reich, M., & Mozuraitis, M. (2015). A cautionary note on the use of the Analysis of Covariance (ANCOVA) in classification designs with and without within-subject factors. *Frontiers in Psychology*, 6. <https://doi.org/10.3389/fpsyg.2015.00474>

Taylor, A. (2011). *Using the GLM Procedure in SPSS*. Sydney, Australia: Macquarie University. Retrieved from <http://www.psy.mq.edu.au/psystat/documents/GLMSPSS.pdf>

Reviewer: 2

Comments to the Author(s)

This is an interesting paper reporting a straightforward study that shows further evidence that pre-pubertal children are aware of the stereotypes associated with voice pitch. My only suggestion is to improve the figures. Violin plots would be a much better way to present the data in the bar charts, for example.

Author's Response to Decision Letter for (RSOS-190656.R0)

See Appendix A.

Decision letter (RSOS-190656.R1)

14-Jun-2019

Dear Dr Cartei,

I am pleased to inform you that your manuscript entitled "Children can control the expression of masculinity and femininity through the voice" is now accepted for publication in Royal Society Open Science.

on behalf of Professor Carolyn McGettigan (Associate Editor) and Essi Viding (Subject Editor)
openscience@royalsociety.org

Appendix A

Dear Editor,

We are very grateful to you and the two reviewers for your very helpful comments and suggestions. Below we provide a detailed response in bold and red.

Kind regards,

the Authors

Editor

I have now received 2 reviews of your paper from relevant experts in the field. I'm happy to report that both reviewers find the study to be of great interest. As you will see, their queries mainly concern some aspects of the reporting of the statistical analyses, presentation of the results, and the availability of the supporting data and code. Please take a look at their requests and address each of them with a revised manuscript - my view is that they constitute a somewhat minor revision to the paper. If any of the requested adjustments to the statistics (e.g. corrections for multiple comparisons) should alter the findings substantially, then please do highlight this in your response letter and discuss accordingly in the revised manuscript.

We thank you for your encouraging comments. Conducting the relevant adjustments did not affect the findings. More details are provided below in our answers to the individual comments by the reviewers.

Reviewer: 1

Comments to the Author(s)

This is a very interesting, well written and coherent article, that directly tackles an important gap in the literature. It reports the results of a study investigating whether children can modulate their voice parameters (namely, F_0 and ΔF) to modulate gender-related attributes when impersonating same-sex child characters. The results suggest that, in fact, both girls and boys lower F_0 and ΔF to masculinise their voices when impersonating stereotypically masculine child characters, and (girls and older boys) raise them for stereotypically feminine same-sex child characters. This is particularly interesting, as the authors mentioned, given the strong sexual dimorphism of the human voice, and the little attention it has received as a marker of the development of gender stereotypes.

The results are sensible, but the manuscript has a few problems (detailed below) that should be addressed before publication. With some modification the manuscript would be appropriate for publication.

General comments:

The main (but solvable) problem I find is related to the statistics. Given that

this is a novel (and somewhat exploratory) study, it is understandable that there is not a priori power (and sample size) analysis. However, this creates a problem for the interpretation of statistical significance, as the existence of an association cannot be inferred simply from p values. In other words, the study may be, for example, underpowered to detect some moderate or weak associations between predictors and outcome variables.

Although overall the results are in line with our hypotheses, we acknowledge the reviewer's point about the potential for underpowered analyses to detect weak effects. The one association that may fall into this is the finding that boys' upward shifts in F0 in the feminine condition did not reach significance compared to the gender-neutral condition. As we explain in the discussion, boys of similar age to the ones in the present study were found to be able to feminise their voices, as previously reported (Cartei et al., 2014). Future studies with a bigger sample size are required to investigate this further, and we have added a statement to this effect in the discussion (lines 233-236):

Both sexes were also capable of raising F0 and ΔF to feminise their voices for the stereotypically feminine child characters, although boys' upward shifts in these acoustic parameters did not reach overall significance (a result that could be further investigated by replicating the study on a larger sample).

In addition, there is no information of influential data points, nor diagnostics of the adequacy model fit (i.e. residual distribution for ANCOVAs and regressions, which I suggest could be included as supplemental material).

We have now added the residual plots (Figures S1 and S2) for the ANCOVAs in the imitation task as supplementary material (lines 158-159). These do not suggest any problems with the assumptions or adequacy of the model.

I would also recommend providing additional information to infer potential associations, like using bootstrap techniques to identify confidence intervals (having CIs would help assessing associations even in the absence of p values). In particular, bootstrapping regressions could really help in dealing with all these issues (see e.g. Fox, 2016, Chapter 21).

We re-run all the regressions with bootstrapping (n=1000) and now we report bootstrapping 95% CI in the text (lines 141-147, 220-222). None of the CIs for the significant regressions include 0.

In addition, there are few descriptive statistics reported, and I think these are important for interpretation and to inform power calculations in future studies and potential replication attempts. I recommend the authors to include a table of descriptives (e.g. n, mean, median, SD, min, max) of F0 and ΔF for the impersonations of the three characters, by participant sex and age.

We have decided to report the table of descriptives (Table S1), as

suggested by the reviewer, in the supplementary material (line 167). Given that age was treated as a continuous variable in the statistical analyses, we used participant year group for Table S1 to give an indication of the age-related changes in the acoustic variables.

Also, according to the review system “It is a condition of publication that authors make their supporting data, code and materials available”, but the link the authors provided to access the data (<http://sro.sussex.ac.uk/id/eprint/83066/>) gives a “404 File not Found” error. Also, I did not see any mention of code (or SPSS syntax, which from the figures I think the authors used) to reproduce the analyses, but because these are clearly described in the manuscript, it is perhaps desirable, but not too necessary in this particular case to include it (other supplementary files are available with the submitted files).

We changed the link to:

<https://sussex.figshare.com/s/285aef9e3bd234a5541c>

The link will become live once the paper is published.

Finally, perhaps due to some sort of problem or confusion with the submission system, or an author oversight, there are no figure captions.

These have now been uploaded.

Specific comments:

Page 2, line 52. The citation should be “Perry, Ohde, & Ashmead, 2001”, instead of “Perry & Ashmead, 2001”.

Corrected.

Page 3, lines 73-76. In the Participants section, the authors categorise the ages of the participant children in years 1 to 6. I wonder why use Years 1-6, instead of age 6-10? I'd rather use the actual age for the sake of simplicity. It is not clear (at least for me) why these categories are used and, more importantly, why use 6 categories when there are only 5 years of age under study. However, if there is a justification behind this decision, I would like to see it in the manuscript, because it is not obvious to me (and probably some readers of the published paper).

We had reported the year groups to show that the children were roughly equally distributed in terms of sex and age across the primary school years. However, we agree with the reviewer that reporting year groups is confusing, so we replaced them with the participants' age range from 6 to 10, which is consistent with the fact that age was entered in the analyses as a continuous covariate. We replaced the original text below (line 74):

24 in UK Years 1&2 (10 girls, mean age=6.48; SE=.14; 14 boys, mean age = 7.0 SE =.11), 22 in Years 3& 4 (12 girls, mean age= 8.6 SE=.13; 10 boys, mean age = 8.4 SE =.17), and 26 in Years 5&6 (14 girls, mean age= 9.97 SE=.14; 12 boys, mean age =9.7 SE =.12)

with:

A total of 72 children (36 girls), aged 6 to 10, took part in this study.

Page 3, line 80. “Participants provided their own verbal consent”. Given that they were children, very young in some cases, shouldn’t it be “assent” instead of “consent”? (see De Lourdes Levy, Larcher, & Kurz, 2003).

Corrected.

Page 4, line 143. For consistency, I think the first results reported in “F0 ($R^2 = .13$, $F(1, 34) = 5.08$, $\beta = -0.36$, $p = .031$, Figure 1a; boys: $R^2 = .21$, $F(1, 34) = 8.89$, $\beta = -.46$, $p = .005$, Figure 1b)” should start with “girls: ”.

Corrected.

Page 5, line 145. Figure 2a cited here, only illustrates the quadratic model, described below without reference to the figure, but not the linear one reported here. Perhaps both the linear and quadratic could be shown in this same figure panel?

Both the linear and quadratic trends (and their R^2 values) have now been added to figure panel 1c.

Page 5, lines 153-154. This is not necessary, but I would personally like to see the significant age and sex interaction here reported illustrated with a figure (or figure panel).

We feel that the interaction is adequately represented by reporting the ΔF by age scatterplots for girls and boys as two adjacent figures (1c, 1d).

Also, Figs 1 and 2 could perhaps be just one fig with 4 (or 5) panels, illustrating all results related to Age and sex differences in the natural voice. **These two figures have now been combined into one (Figure 1), and all the other figures re-numbered accordingly.**

Page 5, lines 159-160. The authors report that the covariate (age) was mean centred. This is the correct decision, but when the interaction is significant, I think the authors should centre to at least low, high and central (age) values to explore the interaction between character type and age (see Delaney & Maxwell, 1981; Schneider, Avivi-Reich, & Mozuraitis, 2015; Taylor, 2011; for an example of this in an analogous design on voice research, see Leongómez, Mileva, Little, & Roberts, 2017). I really do think this will help interpret the interaction, and how the model slopes vary for kids of different ages. In addition, this technique will keep the n as the values are estimated,

and you would not lose statistical power.

Although we had presented participants' age as three separate year groups in the *Participants* section, age was treated as a continuous variable in all the analyses. Therefore, we have now changed the participants section to reflect this (line 73). In the stats we do not believe that mean centering within groups would be relevant.

Page 5, line 163. Did you adjust the alpha for multiple comparisons? An alternative, to limit Type I error inflation, would be to instead use (simple) planned contrasts on the estimated values from the covariate (centred to different values), using the neutral character responses as reference (i.e. comparing responses to the feminine and masculine character vs the neutral). In SPSS, see https://www.ibm.com/support/knowledgecenter/no/SSLVMB_24.0.0/spss/advanced/syn_glm_repeated_measures_contrasts_wsfactor.html

The reported multiple comparisons are all Bonferroni corrected, and we have now mentioned this explicitly in the paper (line 177).

Page 6, line 192. Here I would suggest centring to low, mid, and high age.

This procedure would be relevant for investigating the interaction effect of age with character type on ΔF in boys. However, as pointed out earlier, although we originally reported participants' age as year groups in the *Participants* section, in all statistical analyses age is treated as a continuous variable. Choosing three different cut-off ages for mean-centering to low, mid and high would be somewhat artificial (e.g. choosing Year 1&2, 3&4, 5&6). We preferred exploring this interaction by running a simple regression to test whether the magnitude of differences between feminine and gender-neutral characterisations and masculine and gender-neutral characterisations changes with age. A similar procedure was followed in Cartei and colleagues (2014).

Cartei, V., Cowles, W., Banerjee, R., & Reby, D. (2014). Control of voice gender in pre-pubertal children. *British Journal of Developmental Psychology*, 32(1), 100-106. Ref: <https://doi.org/10.1111/bjdp.12027>

Page 6, lines 210-211. "(...) boys spoke with an overall 4.5% lower ΔF in girls" should be "4.5% lower ΔF than girls".

Corrected (line 234).

Reviewer: 2

Comments to the Author(s)

This is an interesting paper reporting a straightforward study that shows further evidence that pre-pubertal children are aware of the stereotypes associated with voice pitch. My only suggestion is to improve the figures. Violin plots would be a much better way to present the data in the bar charts,

for example.

We created the violin plots, but we did not find that they improved the readability of the results. Therefore, we added them as supplementary material, Figure S3 (lines 178-9).